# Purification and Identification of a Novel Angiotensin Converting Enzyme Inhibitory Peptide from the Enzymatic Hydrolysate of *Lepidotrigla microptera*

**DOI:** 10.3390/foods11131889

**Published:** 2022-06-26

**Authors:** Xuejia Hu, Zhiyuan Dai, Renyao Jin

**Affiliations:** Institute of Seafood, Zhejiang Gongshang University, Hangzhou 310035, China; huxj66666@163.com (X.H.); nodjin@163.com (R.J.)

**Keywords:** ACE inhibitory peptide, *Lepidotrigla microptera*, identification, molecular docking, stability

## Abstract

In this study, *Lepidotrigla microptera* were hydrolyzed with four different proteolytic enzymes (Papain, neutrase, flavourzyme, and alcalase), and their distribution of molecular weights and ACE-inhibitory activity were tested. The alcalase hydrolysates showed the maximum ACE-inhibitory activity. A novel ACE-inhibitory peptide was isolated and purified from *Lepidotrigla microptera* protein hydrolysate (LMPH) using ultrafiltration, gel filtration chromatography, and preparative high performance liquid chromatography (prep-HPLC). The amino acid sequence of the purified peptide was identified as Phe-Leu-Thr-Ala-Gly-Leu-Leu-Asp (DLTAGLLE), and the IC_50_ value was 0.13 mg/mL. The ACE-inhibitory activity of DLTAGLLE was stable across a range of temperatures (<100 °C) and pH values (3.0–11.0) and retained after gastrointestinal digestion. DLTAGLLE was further identified as a noncompetitive inhibitor by Lineweaver–Burk plot. The molecular docking simulation showed that DLTAGLLE showed a high binding affinity with ACE sites by seven short hydrogen bonds. As the first reported antihypertensive peptide extracted from alcalase hydrolysate of *Lepidotrigla microptera,* DLTAGLLE has the potential to develop functional food or novel ACE-inhibitor drugs.

## 1. Introduction

In recent years, long-standing stress, unhealthy dietary behavior, and fierce competition have compounded the problems of hypertension for many people. A report in 2005 projected that the number of adults with high blood pressure is expected to rise by about 60% to a total of 1.56 billion (1.54 billion to 1.58 billion) in 2025 [1,2]. Therefore, hypertension has become a major public-health challenge worldwide. Hypertension is a condition of high blood pressure that occurs when the arteriolar resistance increases in blood vessels as a result of increased peripheral arteriolar smooth muscle tone [3]. Central to the pathology of hypertension and its complications is the renin-angiotensin-aldosterone system (RAAS), and the principal component of the RAAS is the angiotensin I-converting enzyme (ACE) [4]. ACE plays dual roles in regulating blood pressure by catalyzing the conversion of angiotensin I to angiotensin II as well as by triggering the inactivation of bradykinin, a vasodilatory peptide [5,6]. Hence, inactivation or inhibition of ACE is recognized as a direct method for the reduction of blood pressure.

In order to treat hypertension effectively [7], many synthetic ACE inhibitors, including ramipril, enalapril, captopril, and others, have been used via the biochemical pathway that leads to vasodilation [8]. Although these are highly effective, they are associated with side effects such as renal impairment, loss of taste, cough, and angioneurotic edema [9,10,11]. Moreover, long-term use of these compounds has been shown to result in aldosterone escape, decreasing their effectiveness [12]. Compared to synthetic drugs, ACE-inhibitory peptides originating from natural proteins are considered to be milder and safer [13]. Furthermore, these peptides usually not only have multifunctional properties but are easily absorbed by humans [14]. Therefore, peptides derived from food proteins caused widespread interest among scientists. Edible proteins from plant or animal sources are a focus for the development of novel ACE-inhibitory peptides. Up to now, active peptides with ACE-inhibitory activity have been released from honeybee pupae [15], cassia obtusifolia seeds [16], and red alga [17].

Marine organisms are considered as important targets for the production of bioactive peptides due to their abundant protein content. In recent years, ACE-inhibitory peptides derived from various marine organisms such as yellowfin tuna [18], squid [19], pipefish [20], and cuttlefish [21] have been extensively reported. *Lepidotrigla microptera* is distributed in the Pacific Northwest and is cheap, high-yield, high in protein, and balanced in amino acids, making it a perfect source of nutrients. However, with its small size, extremely muddy flavor, and large quantity of fish bones, *Lepidotrigla microptera* is a low- market-value resource, and studies on bioactive polypeptide obtained from *Lepidotrigla microptera* protein (LMP) are scarce [22]. Bioactive peptides can be prepared by chemical hydrolysis, enzymatic hydrolysis, or microbial fermentation [23]. Among them, enzymatic hydrolysis is the most commonly used method because it is safe, high in product yields, generates much less waste, and consumes less energy [24]. The bioavailability and functional activity of peptides are closely related to the site and degree of protein of the hydrolysis.

Therefore, in this study, *Lepidotrigla microptera* was used as raw material, and ACE inhibition rate and molecular weight distribution were selected as indicators to screen the best protease. The component with the most potent activity was isolated and purified by ultrafiltration, gel filtration chromatography, and prep-HPLC; then the amino acid sequence of the purified peptides was identified. Furthermore, Lineweaver–Burk plot and molecular docking were performed to demonstrate the ACE-inhibitory mechanism of active peptide. Finally, the stability of the ACE-inhibitory peptide was further studied. This study isolated and identified a new ACE inhibitor derived from LMPH that is promising as a natural alternative to synthetic hypertension drugs. Meanwhile, the study provided data support and a theoretical foundation for diversified development and utilization of *Lepidotrigla microptera*.

## 2. Materials and Methods

### 2.1. Materials

The Fresh *Lepidotrigla microptera* was provided by Anxin Products Co., Ltd. (Zhoushan, China). Hippuric acid (HA), hippuryl-histidyl-leucine (HHL), Papain (20,000 U/mg), Neutrase (80,000 U/mg), Flavourzyme (60,000 U/mg), and Alcalase (100,000 U/mg) were purchased from Solarbio Technology Co., (Beijing, China). All other reagents used in this study were reagent-grade chemicals.

### 2.2. Preparation of LMPH

Freeze-dried samples of *Lepidotrigla microptera* meat were comminuted in a grinder then the sample powder passed through a 200-mesh sieve. Referring to previous studies, papain, neutrase, flavourzyme, and alcalase are frequently used in the production of ACE-inhibitory peptides [25,26]. Different proteases have different restriction sites, so that peptides of different types, sizes, and biological activities were produced [24]. Therefore, in this study, four enzymes were compared to select the best hydrolytic enzyme for the production of ACE-inhibitory peptide. The treated sample powder was mixed with deionized water at the ratio of 1:10 (*w*/*v*). Papain (pH 6.5, 50 °C), neutrase (pH 7.0, 48 °C), flavourzyme (pH 7.1, 50 °C), and alcalase (pH 9.0, 55 °C) were added to four sample solutions with suitable temperature and pH, respectively, at a ratio of 5000 U/g. During the reaction, 1M NaOH and 1M HCl were used to adjust the pH of the mixture. After inactivation at 95 °C for 10 min, the reaction mixture was centrifuged at 5000 rpm, 4 °C for 15 min. The supernatant was freeze-dried for further analysis and was designated as the *Lepidotrigla microptera* protein hydrolysates (LMPH).

### 2.3. Determination of the Hydrolysis Degree (DH)

The DH were determined according to the pH-stat method of Marquez [27] with the following modifications.

### 2.4. Measurement of ACE-Inhibitory Activity

ACE-inhibitory activity was determined following the method of Cushman [27]. The reaction system comprised a sample (10 μL), HHL (2.5 mM, 30 μL), and ACE solution (0.1 U/mL, 20 μL) in sodium borate buffer (pH 8.3, 0.3 M NaCl). The mixture of HHL and the sample was pre-incubated for 15 min at 37 °C before adding the ACE solution. The reaction was carried out for 40 min at 37 °C in a water bath and terminated by adding HCl (1 mol/L, 60 μL). Then 20 μL of the reaction mixture was injected into an HPLC system (Shimadzu, Japan) equipped with a C-18 column (150 × 4.6 mm, 5 μm). In order to detect the concentration of HA (hippuric acid), isocratic elution was performed for 25 min in 80% mobile phase A (0.05% Trifluoroacetic acid in H_2_O) and 20% mobile phase B (Acetonitrile) at a 0.8 mL/min flow rate, and the detection wavelength was 280 nm. The ACE-inhibitory activity was calculated according to the formula:ACE-inhibitory activity (%) = (A1 − A2)/A1(1)
where A1 and A2 equal the peak areas of blank control and the sample, respectively.

### 2.5. Distribution of Molecular Weights

LMPH were analyzed for their relative molecular weight distribution using a high-performance liquid chromatography system (HPLC; Waters e2686, Milford, MA, USA) with a TSK gel 2000 SWXL (7.8 × 300 mm) column with some modification, as described by Li et al. [28] and Xie et al. [29]. The following standards were used to acquire a molecular weight calibration curve: The hippuryl-his-leu (429 Da), aprotinin (6500 Da), cytochrome c (12,400 Da), and carbonic anhydrase (29,000 Da).

### 2.6. Fractionation by Ultrafiltration

The LMPH of alcalase were dissolved in deionized water (30 g/L) and filtered sequentially using an ultrafiltration unit through four hyperfiltration membranes with a molecular weight (MW) cutoff of 10, 5, 3, and 1 kDa, respectively. Five fractions were collected, namely as LMPH I (MW < 1 kDa), LMPH II (MW 1–3 kDa), LMPH III (MW 3–5 kDa), LMPH IV (MW 5–10 kDa), and LMPH V (MW > 10 kDa). Each part was freeze-dried for measuring ACE-inhibitory activity. The component with the highest ACE-inhibitory activity was stored at −70 °C until use.

### 2.7. Gel Filtration Chromatography

The most active fraction was further separated by Sephadex G-15 column (1.5 × 80 cm) according to the method of Wu et al. [30]. The lyophilized sample (50 g) was suspended in 1 L of deionized water then loaded to a Sephadex G-15, which was eluted with deionized water at a flow rate of 1.5 mL/min. Fractions (5 mL each per tube) were collected and monitored at 220 nm. to obtain the elution curves. Each fraction was tested for ACE-inhibitory activity. The fraction exhibiting the most potent ACE-inhibitory activity was freeze-dried and stored for further study.

### 2.8. Purification of ACE Inhibitor by Prep-HPLC

The fraction with the maximum ACE-inhibitory activity after gel filtration chromatography was further separated by prep-HPLC on a C18 column at a flow rate of 1.0 mL/min. Separation was performed under a linear gradient of acetonitrile (5–45%) that contained 0.1% trifluoroacetic acid. The absorbance was assayed at 220 nm. Each fraction was collected and measured for ACE-inhibitory activity. The fraction showing the highest ACE-inhibitory activity was freeze-dried and designated as ACE-inhibitory peptides.

### 2.9. Peptide Identification by LC–MS/MS

The most active fraction obtained from the RP-HPLC was analyzed by Waters Synapt HDMS system (Waters, Milford, MA, USA) on an RP-C18 column (150 × 0.15 mm) with a flow rate of 0.3 mL/min. The column temperature was 30 °C and the detection wavelength was 220 nm. The mobile phase consisted of acetonitrile with 0.1% formic acid (mobile phase A) and 0.1% formic acid aqueous solution (mobile phase B). The following linear gradient was used: 0–50 min, 4% A; 50–54 min, 50% A; 54–60 min, 100% A. For mass spectrometry, parameters were set as follows: collision energy, 6 eV; ion source temperature, 100 °C; detector voltage, 1800 V; mass range, 300–2000 *m*/*z*.

### 2.10. Peptide Synthesis

The peptides with the sequence NSSRFGKF, DLTAGLLE, and YLTLFLLT were synthesized at Bank Peptide Biological Technology Co., Ltd. (Hefei, China).

### 2.11. Inhibitory Kinetics Study

We used Lineweaver–Burk plots to investigate the ACE-inhibitory pattern of the most potent peptide. Different concentrations of ACE-inhibitory peptide (0, 60, and 120 μM) were mixed with various concentrations of HHL solution (1, 2, 3, 4 and 5 mM) and pre-incubated for 5 min at 37 °C. Then the reaction was initiated by adding 80 µL of ACE and terminated by adding 20 µL of 1 mol/L HCl. The Lineweaver–Burk plot of the reciprocal of the HHL concentration (1/[s]) and the HA production rate (1/[v]) were used to analyze the inhibitory kinetics of peptide.

### 2.12. Stability of ACE Inhibitory Activity

#### 2.12.1. Temperature

The sample (0.2 mg/mL) was treated at various temperatures (20, 40, 60, 80, and 100 °C) for 1 h. After cooling the solution to indoor temperature (25 °C), the ACE inhibition activity was detected.

#### 2.12.2. pH

The ACE-inhibitory peptide solution (0.2 mg/mL) was incubated at different pH levels (2, 4, 6, 8, and 10) at 37 °C for 2 h, and the pH value was adjusted by adding HCl (1 mol/L) and NaOH (1 mol/L). Before the ACE inhibition rate was determined, the pH value was adjusted to 7.0.

#### 2.12.3. Simulated Gastrointestinal Digestion

Simulated gastrointestinal digestion of ACE-inhibitory peptides was conducted following the method of Lai et al. [31] with slight modifications. In a nutshell, peptides (0.2 mg/mL, 1 mL) were digested by pepsin (0.05%, 1 mL) for 2 h at 37 °C, and the pH value was adjusted to 2.0 by HCl of 1 mol/L. Subsequently, the digest was adjusted to pH 7.5 using NaOH solution (1 mol/L), with half of the digest collected as peptic digest and the other half further digested by pancreatin (0.05%, 1 mL) for another 2 h at 37 °C. Finally, the digestion solution was kept in 95 °C for 15 min to terminate the reaction.

### 2.13. Molecular Docking

The three-dimensional crystal structure of ACE (PDB: 1O8A) was imported from the Protein Data Bank. Prior to docking analysis, the water molecules and inhibitor lisinopril were removed while the chloride atoms and cofactors zinc were reserved in the ACE model. The polar hydrogens were then added to the ACE crystal structure. Based on molecular simulations, the ACE was selected as the macromolecular receptor. The three-dimensional structure of the peptide was constructed using PyMol software, which was selected as small-molecule ligands. Molecular docking between the peptide molecule and ACE was carried out using a flexible docking tool via AutoDockTools-1.5.6 software. The docking runs were performed with a radius of 126 Å with coordinates x: 40.586, y: 37.374, and z: 43.446. The optimum docking position of the peptide in complex with ACE was selected according to the binding energy value and scores.

## 3. Results and Discussion

### 3.1. Distribution of Molecular Weights and ACE-Inhibitory Activity of Hydrolysates

The extent of protein degradation by protease was estimated by assessing the DH [32]. The order from high to low of the DH of the protease hydrolysates was alcalase hydrolysate (51.16%), flavourzyme hydrolysate (36.22%), neutrase hydrolysates (28.37%), and papain (21.26%). The different DH values of the four protease hydrolysates indicated that the specificity of the four enzymes was different, leading to different ACE-inhibitory activities. ACE-inhibitory activity of the LMPH from various enzymes is listed in Table 1. The ACE inhibition rate of alcalase hydrolysate IC_50_ (5.7 ± 0.15 mg/mL) was the lowest, followed by neutrase hydrolysates IC_50_ (6.5 ± 0.13 mg/mL) and flavourzyme hydrolysate IC_50_ (6.1 ± 0.11 mg/mL); the ACE inhibition rate of papain IC_50_ (6.8 ± 0.16 mg/mL) was the highest.

The ACE inhibition rate of the peptides was strongly linked to their molecular weight [15]. Low-molecular-weight peptides are more impressionable and have a critical effect in their ACE-inhibitory activity because they are conducive to combination with the target molecule. The molecular weight distributions of standards are shown in Figure 1a with a regression equation of y = −0.2025x + 6.8751, R^2^ = 0.9758. Additionally, the distribution of the molecular weight of LMPH hydrolyzed by four proteolytic enzymes was presented in Figure 1b. The proportion of peptides with molecular weights < 1000 Da hydrolyzed by papain, neutrase, flavourzyme, and alcalase was 33.29%, 59.11%, 65.91%, and 88.21%, respectively. The control group had the lowest ratio of components < 1000 Da (8.94%). Lee et al. [33] found that peptides with a lower molecular weight seem to exhibit higher ACE-inhibitory activity compared to high-molecular-weight peptides because peptides with lower molecular weight were conducive to combination with the target molecule. Dae et al. reported that peptides with smaller molecular weight from T. giganteum had higher ACE-inhibitory activity [34]. Wu et al. showed that peptides with smaller size (<2 kDa) prepared from red algae showed the greatest ACE-inhibitory activity [17]. The findings were consistent with the results of this study. Thus, compared to the other three proteases, alcalase was more effective at hydrolyzing LMP. Therefore, we chose alcalase for subsequent research.

### 3.2. Ultrafiltration of LMPH

After the LMP was hydrolyzed with alcalase, ultrafiltration helped enrich the micromolecules and remove macromolecules as well as improve ACE-inhibitory activity. Through ultrafiltration, the hydrolysate was divided into five different fractions, namely LMPH I (MW < 1 kDa), LMPH II (MW 1–3 kDa), LMPH III (MW 3–5 kDa), LMPH IV (MW 5–10 kDa), and LMPH V (MW > 10 kDa); then it was lyophilized and compared in terms of the ACE-inhibitory activity of each proportion with different molecular weight ranges. The molecular weight of different fractions was inversely proportional to the ACE activity; that is to say, the ACE-inhibition rate decreased with the increase in molecular weight. The IC_50_ values of LMPH I, LMPH II, LMPH III, LMPH IV, and LMPH V were 0.77 ± 0.04, 1.23 ± 0.07, 1.8 ± 0.05, 4.31 ± 0.03, and 6.11 ± 0.06 mg/mL, respectively. The LMPH I fraction showed the highest inhibition ratio on ACE, which was significantly higher than that of LMPH II, LMPH III, LMPH IV, and LMPH V. Thus, LMPH I was collected for further purification.

### 3.3. Gel Filtration Chromatography of LMPH I Fraction

The LMPH I was separated ulteriorly by a Sephadex G-15 chromatography column, and five fractions (F1, F2, F3, F4, and F5) were subsequently eluted from the column according to molecular size (Figure 2a). Fractions F4 and F5 showed the most potent ACE-inhibitory activity, which exhibited IC_50_ values of 0.38 ± 0.03 and 0.41 ± 0.05 mg/mL, respectively, and there was no significant difference between the two fractions (Figure 2b). Considering that the F4 part had not only the highest ACE-inhibitory activity but also the highest yield, the F4 fraction was selected for further purification.

### 3.4. Separation of the F4 Fraction by Prep-HPLC

The F4 from gel filtration chromatography was further separated by prep -HPLC into 19 fractions (named from F4-1 to F4-19) (Figure 3a), which were collected for ACE-inhibitory assays (Figure 3b). Among the 19 fractions, F4-12 (with a retention time of 20.1–21.5 min) exhibited the most potent ACE-inhibitory activity, with an IC_50_ value of 0.28 mg/mL. Usually, separations that used prep -HPLC were based on hydrophilic properties; thus, hydrophilic peptides were eluted first and hydrophobic peptides were eluted later [35]. F4-12 was collected at the middle phase of prep -HPLC; hence, the content of hydrophobic amino acids was higher [36,37] (Figure 3a). Rui et al. found that most ACE-inhibitory peptides contained more hydrophobic amino acids, which suggested that ACE-inhibitory peptides were obtained by elution in the middle or end of pre-HPLC operation [38]. Tao et al. reported that ten main fractions (namely as HP-1 to HP-10) were separated by the order of elution, among which the HP-6 proved the most potent to inhibit the activity of ACE with a value of 76.27 ± 0.11% [39]. Thus, the results obtained from F2-12 were in accordance with these studies. F2-12 was collected for further characterization studies.

### 3.5. Identification of Peptides from LMPH

The amino acid sequence of F4-12 was analyzed by UPLC-MS/MS. As presented in Figure 4, the amino acid sequence of F4-12 was identified as Asn-Ser-Ser-Arg-Phe-Gly-Lys-Phe (NSSRFGKF), Phe-Leu-Thr-Ala-Gly-Leu-Leu-Asp (DLTAGLLE), and Tyr-Leu-Thr-Leu-Phe-Leu-Leu-Thr (YLTLFLLT). Subsequently, the three peptides were synthesized chemically and the IC_50_ values were measured. The NSSRFGKF showed no ACE-inhibitory activity; the other two peptides, DLTAGLLE and YLTLFLLT, had IC_50_ values of 0.13 and 0.21 mg/mL, respectively. DLTAGLLE showed the highest ACE inhibitory activity, and the DLTAGLLE peptide originating from LMP had not been reported previously. In addition, no record of this peptide is found by searching the BIOPEP database.

According to the activity–structure correlations in peptides, the occurrence of hydrophobic (Leu, Ile, Ala, Val) and/or aromatic (Tyr, Trp, Phe) at the first three C-terminal positions can dramatically increase the binding affinity between the peptide and ACE [30,40,41]. Rafifik et al. [26] isolated and purified novel ACE-inhibitory peptides (Val-Glu-LEu-Tyr-Pro) from the muscle of cuttlefish which contained hydrophobic amino acids at the C-terminal. In this perspective, DLTAGLLE had hydrophobic amino acids at the first three C-terminal positions, which meet the structural requirements for ACE inhibition. In addition, another important factor affecting ACE-inhibitory activity was the proportion of hydrophobic amino acids to hydrophilic amino acids in the peptide chain. The ACE inhibition rate increased with the increase in the proportion of hydrophobic amino acids. Maria et al. [42] reported that the ACE-inhibitory peptides from casein hydrolysate had low molecular weight and hydrophobic properties. Mirzaei et al. prepared ACE-inhibitory peptide (LPESVHLDK) from yeast protein; the ratio of hydrophobic amino acids was 44.44% [43], whereas DLTAGLLE had a higher percentage of hydrophobic amino acids (50%). Another vital parameter for ACE inhibition is the length of the peptides. It has been shown that ACE-inhibitory peptides usually contain 2–12 amino acids, which can move into the bloodstream without losing their properties [44].

### 3.6. ACE Inhibition Mode and DLTAGLLE Stability

To further demonstrate the inhibitory kinetic mechanism of the novel ACE-inhibitory peptide, the ACE inhibition patterns of DLTAGLLE were estimated by Lineweaver–Burk plot analysis. The results are as shown in Figure 5. The maximum velocity (Vmax) of the enzyme reaction without the purified DLTAGLLE peptide was 20 mM/min; nevertheless, Vmax was indicated to be 5 mM/min and 2.78 mM/min at 60 μM and 120 μM DLTAGLLE, respectively. The Michaelis–Menten constant (Km) of the reaction was 4.1 mM. The Vmax decreased and the Km remained unchanged as the peptide concentration increased. The result exhibited that the inhibitory kinetic mechanism of DLTAGLLE peptide was noncompetitive, which indicates that both substrate and inhibitor can bind to the enzyme to produce terminal complexes, whether or not the substrate molecule binds. Thus, the DLTAGLLE peptide binds to a diverse site from the substrate. As a result, the DLTAGLLE acts as an ACE inhibitor by forming an enzyme–inhibitor or enzyme–substrate–inhibitor complex during the reaction process, resulting in a decrease in catalytic efficiency [45]. Noncompetitive inhibitors have also been found in tuna [25], oyster [46], and cuttlefish [21].

Figure 6a gave the thermal stability of the DLTAGLLE peptide. In the temperature range of 20 to 60 °C, the inhibitory rate of peptide on ACE fluctuated slightly with increasing temperature but remained relatively stable. However, with a temperature above 60 °C, the activity of DLTAGLLE peptide slightly decreased. The reason may be that the sequence and conformation of peptides changed due to the increase in temperature, resulting in a decline in ACE-inhibitory activity. Overall, the DLTAGLLE peptide maintained high ACE-inhibitory activity at different temperatures and had satisfactory thermal stability. Rafifik et al. [26] reported that the purified peptide was heat-stable at 4–100 °C. Hwang et al. [47] reported that different heating temperatures (20–100 °C) had no significant effect on the activity of ACE-inhibitory peptides extracted from tuna cooking juice.

The ACE-inhibitory activity of DLTAGLLE peptide under different pH treatments is shown in Figure 6b. With the increase in pH from 3 to 11, the inhibitory rate of DLTAGLLE peptide on ACE first increased and then decreased. The results indicate that the DLTAGLLE peptide would cause deamidation and racemization under alkaline conditions, which changes the structure of DLTAGLLE and affects its activity. In conclusion, the DLTAGLLE peptide maintained high ACE-inhibitory activity at all pH values.

The results in Figure 6c show that the digestive stability of the DLTAGLLE peptide reflects the digestion process of DLTAGLLE peptide in the body. It is indicated that the relative ACE inhibition activity of the DLTAGLLE peptide was essentially unchanged before and after digestion. Thus, the peptide possessed strong digestion resistance. Zhang et al. [48] discovered that the active sequence of the peptides (ASPYAFGL) could not be destroyed by gastrointestinal protease, which indicated that the peptide (ASPYAFGL) had strong anti-enzyme properties.

### 3.7. Molecular Docking

Molecular docking is an effective way to predict the interactions between the receptor and ligand by calculating the binding sites, the interaction energies, and other interaction information. ACE- and DLTAGLLE-inhibitory peptide interaction may be analyzed by molecular docking. Figure 7a shows the structure of the DLTAGLLE peptide, and Figure 7b shows the docking diagram of DLTAGLLE binding to ACE. The results show that the DLTAGLLE peptide has an extended conformation and is deeply embedded in the three-dimensional structure of ACE. Figure 7c reveals the interaction between DLTAGLLE and the residues of ACE; the green sticks indicate DLTAGLLE peptide, the cyan sticks indicate the amino acid residues of the ACE sites, and the yellow dotted line indicates the interaction between the sites of ACE and DLTAGLLE. The result reveals that DLTAGLLE successfully docked with ACE residues and displayed twelve hydrogen bonds with SER517, GLU123, THR78, ASP358, ARG522, and GLU403 of ACE. These hydrogen bonds were short, ranging from 1.9 Å to 3.0 Å. The existence of these short hydrogen bonds demonstrated that DLTAGLLE has strong binding ability to ACE. DLTAGLLE combined well with ACE, thus bringing down the ACE activity.

Pina et al. reported that the active site of ACE involves 12 critical amino acid residues, namely Gln281, Ala354, His513, His383, Glu411, Glu162, His353, Glu384, His387, Lys511, Tyr523, and Tyr520 [49]. In this study, the DLTAGLLE -bound sites SER517, GLU123, THR78, ASP358, ARG522, and GLU403 were not key amino acid residues of ACE. Hence, a noncompetitive interplay was manifested between DLTAGLLE and ACE. The result was in accord with the ACE-inhibitory kinetic of DLTAGLLE measured by Lineweaver–Burk plots.

## 4. Conclusions

*Lepidotrigla microptera* protein hydrolysates produced from papain, neutrase, flavourzyme, and alcalase showed substantial improvement in inhibitory activity ACE compared to *Lepidotrigla microptera* materials. Among the results, alcalase hydrolysates exhibited the highest inhibiting activity on ACE. A novel ACE-inhibitory peptide, DLTAGLLE, was purified from the *Lepidotrigla microptera* protein hydrolysates using ultrafiltration, G-15 gel chromatography, and prep-HPLC. DLTAGLLE was further proved to have noncompetitive inhibitory characteristics and revealed good stability to temperature, pH, and simulated gastrointestinal digestion. The molecular docking revealed that DLTAGLLE had a high binding force with ACE sites using short hydrogen bonds. The results of this study implied that DLTAGLLE has broad prospects in application to functional foods or therapeutic agents against hypertension. However, further research is needed to assess the safety of DLTAGLLE, including the absence of toxicity, allergenicity, and cytotoxicity; the antihypertensive mechanism of DLTAGLLE in vivo also needs further research.

## Figures and Tables

**Figure 1 foods-11-01889-f001:**
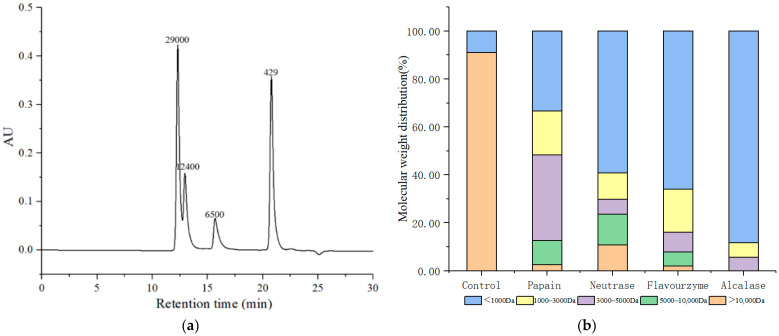
Molecular mass distribution of standards (**a**), and molecular mass distribution of LMPH obtained by papain, neutrase, flavourzyme, and alcalase (**b**).

**Figure 2 foods-11-01889-f002:**
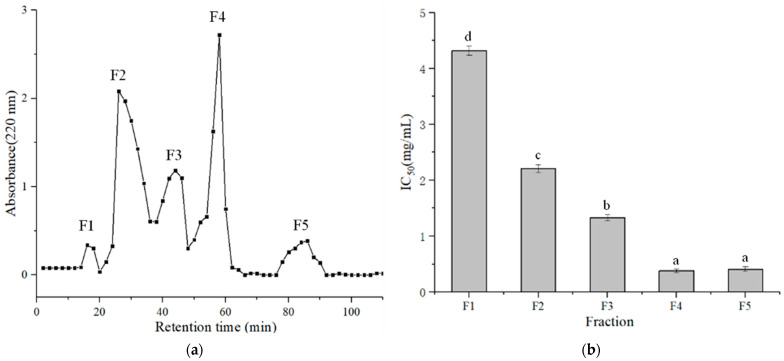
Gel filtration chromatography of LMPH I on a Sephadex G-15 column (**a**), and ACE-inhibitory activities of fractions F1–F5 (**b**). The column values of a–d with the different superscript indicated a significant difference (*p* < 0.05).

**Figure 3 foods-11-01889-f003:**
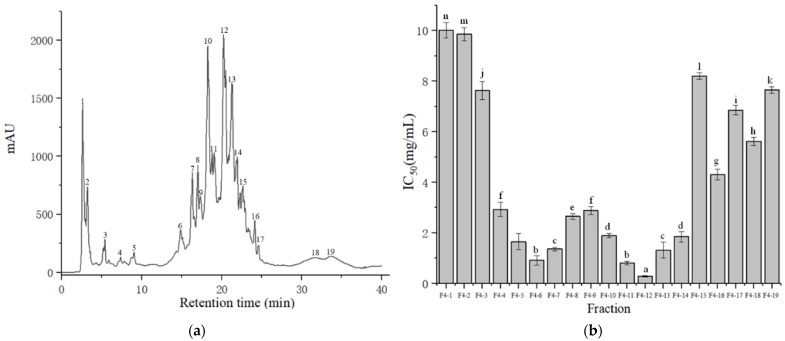
Chromatograph obtained by prep-HPLC of fraction F4 (**a**) and ACE inhibitory activities of fractions F4-1–F4-19 (**b**). The column values of a–n with the different superscript indicated a significant difference (*p* < 0.05).

**Figure 4 foods-11-01889-f004:**
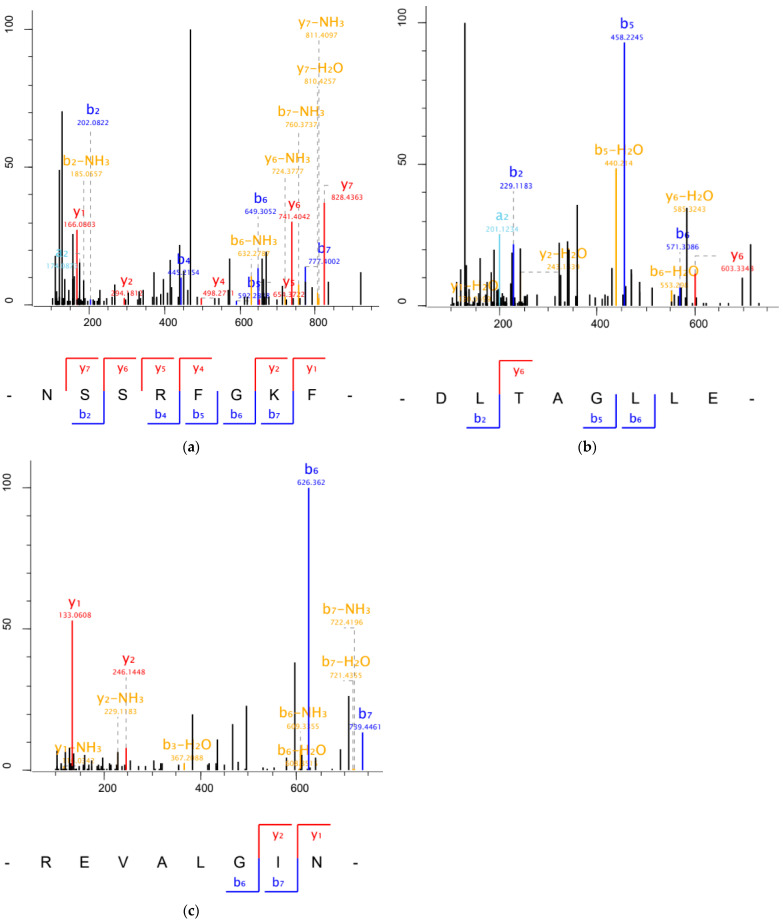
Molecular masses and amino acid sequences of ACE-inhibitory peptides from the purified F4-12 fraction. MS/MS spectra of NSSRFGKF (**a**), DLTAGLLE (**b**), and REVALGIN (**c**).

**Figure 5 foods-11-01889-f005:**
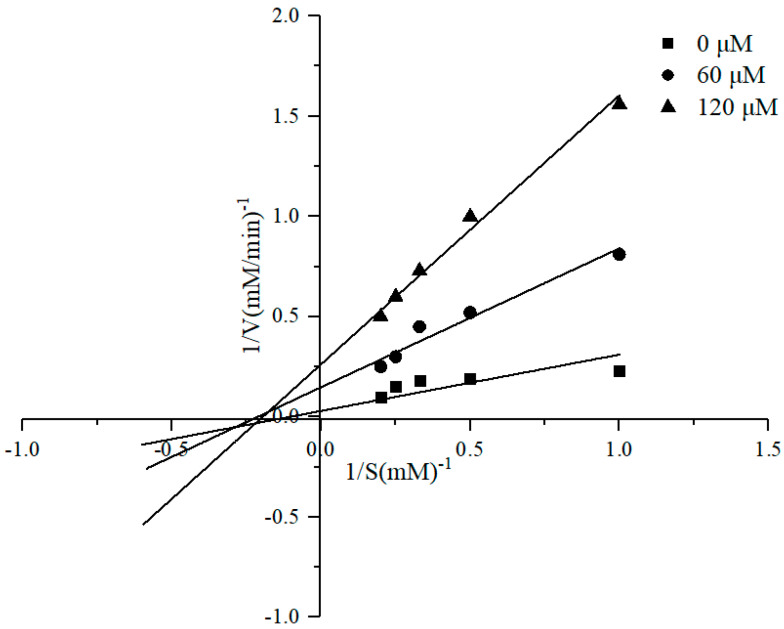
Lineweaver−Burk plots of ACE inhibition by the DLTAGLLE peptide.

**Figure 6 foods-11-01889-f006:**
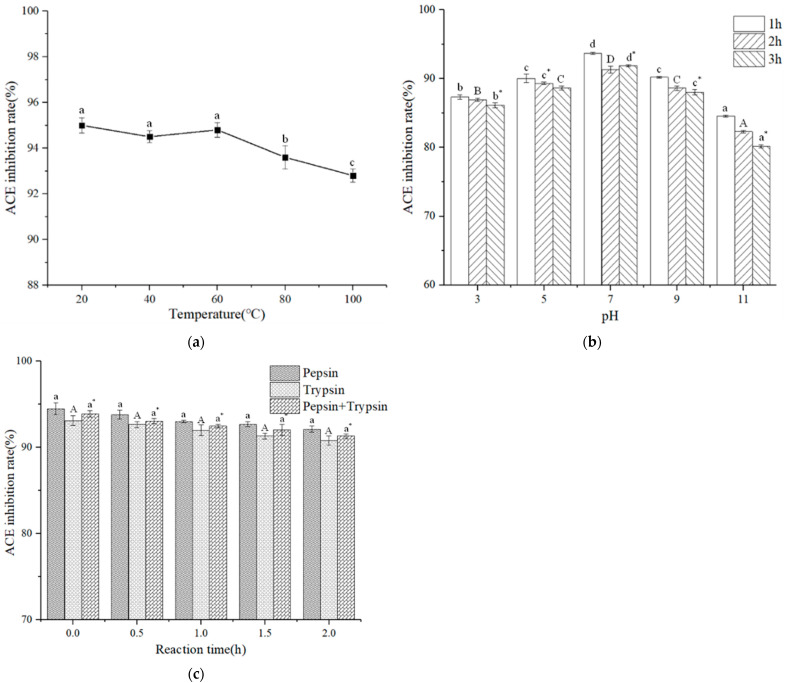
Stability of the DLTAGLLE. Effect of temperature (**a**), pH (**b**), and gastrointestinal digestion (**c**) on the ACE inhibitory activity of DLTAGLLE. The column values of A–D, a–d and a*–d* with the different superscript indicated a significant difference (*p* < 0.05).

**Figure 7 foods-11-01889-f007:**
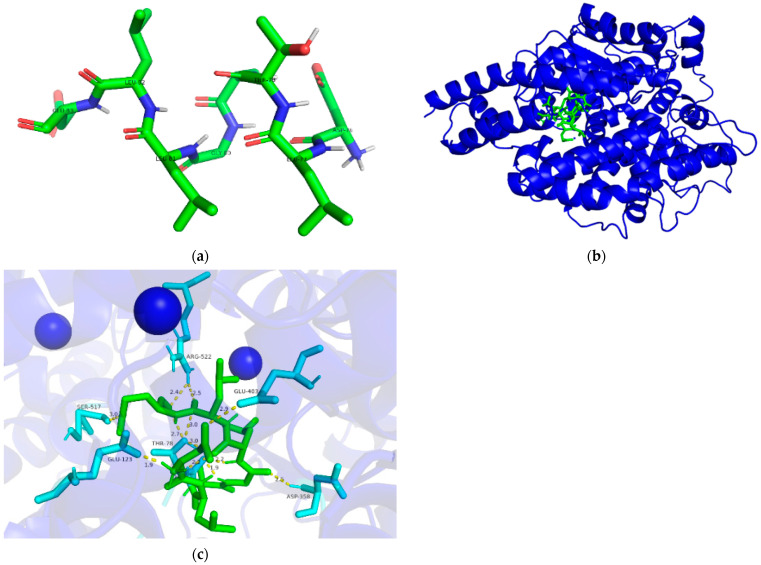
Molecular docking simulation of DLTAGLLE binding to ACE. (**a**) Structure of the peptide. (**b**) Docking diagram of DLTAGLLE (green) binding to ACE (shown as cartoon). (**c**) Interaction between DLTAGLLE (green) and residues of ACE (shown as sticks).

**Table 1 foods-11-01889-t001:** ACE-inhibitory activity of LMPH.

Group	Protein Concentration (mg/mL)				IC_50_ (mg/mL)
	1	3	5	7	
Papain	28.4 ± 1.6 ^b^	32.61 ± 0.9 ^b^	45.12 ± 1.9 ^a^	55.12 ± 2.3 ^a^	6.8 ± 0.16 ^d^
Neutrase	10.23 ± 2.3 ^a^	28.31 ± 1.3 ^a^	48.13 ± 0.8 ^c^	59.23 ± 1.2 ^b^	6.5 ± 0.13 ^c^
Alcalase	32.12 ± 2.5 ^d^	38.21 ± 1.1 ^d^	46.51 ± 1.1 ^b^	63.62 ± 1.4 ^d^	5.7 ± 0.15 ^a^
Flavourzyme	29.41 ± 0.8 ^c^	35.15 ± 2.6 ^c^	49.35 ± 1.5 ^d^	61.33 ± 1.3 ^c^	6.1 ± 0.11 ^b^

Values represent the means ± standard error (*n* = 3); different letters (a–d) in the same line indicate significant differences at *p* < 0.05.

## Data Availability

The authors confirm that the data supporting the findings of this study are available within the article.

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
