# Peer review of "Purification and Identification of a Novel Angiotensin Converting Enzyme Inhibitory Peptide from the Enzymatic Hydrolysate of Lepidotrigla microptera"

_foods, 2022, doi:10.3390/foods11131889_

Round 1
Reviewer 1 Report
The present work describes the isolation and characterization of a peptide with inhibitory activity on ACE from Lepidotrigla microptera protein hydrolysate (LMPH). LMPH were produced after incubation with four different proteolytic enzymes (papain, neutrase, flavourzyme and alcalase), and the hydrosylate generated by the action of the alcalase enzyme was the one that exhibited the greatest inhibitory activity on ACE. The alcalase hydrosylate was submitted to several chromatographic steps and resulted in the isolation and characterization of a novel ACE inhibitory peptide (DLTAGLLE). The introduction is adequate. The methodology was properly carried out. The results are interesting and presented clearly. The article has scientific merit and is within the scope of the journal.
I suggest that some adjustments be made to lines 40, 205-206, and Figure 2a (with the identification of each chromatographic fraction).
Author Response
Answer: According to the opinion of reviewer, we have modified the sentence (for details see line 40, 227-228), meanwhile, we also made some adjustments to Figure 2.

Reviewer 2 Report
This is an interesting well-written manuscript dealing with bioactive ACE-inhibitory peptides. The manuscript can be published after some modifications.
L51: in this part please at fires mention the other sources of ACE-inhibitory peptides, for example: Biologically active peptides with ACE-inhibitory activity can be released from different sources including plant and animal-based proteins.
L54-60: need a reference.
I think the authors should add a section to the introduction regarding the methods used to prepare bioactive peptides as well as the parameters affecting their bioactivity such as the sequence, time of hydrolysis, degree of hydrolysis, …
Why did you select these enzymes (Papain, neutrase, flavourzyme and alcalase) for your research? Was there a specific reason?
L80-88: Is this method of hydrolysate production the best method? Because we know that the activities of these enzymes are different and we can predict the results! I think that it was better to work based on the degree of hydrolysis (DH) instead of the time of hydrolysis. Also, I think that the DH determination can help you for a better discussion.
L161-164: What were these temperatures to simulate?
The discussion part is generally weak. Only results have been reported. No comparison has been made with the results of other studies at all. Therefore, the discussion part of the manuscript should be improved.
L349: Table caption: different letters in the same row or column?
L359-360: in figure 2 caption the F1-F5 should be defined.
Author Response
This is an interesting well-written manuscript dealing with bioactive ACE-inhibitory peptides. The manuscript can be published after some modifications.
L51: in this part please at fires mention the other sources of ACE-inhibitory peptides, for example: Biologically active peptides with ACE-inhibitory activity can be released from different sources including plant and animal-based proteins.
Answer: we have added the other sources of ACE-inhibitory peptides, for example: honeybee pupae, cassia obtusifolia seeds and red alga (for details see line 51-52).
L54-60: need a reference.
I think the authors should add a section to the introduction regarding the methods used to prepare bioactive peptides as well as the parameters affecting their bioactivity such as the sequence, time of hydrolysis, degree of hydrolysis.
Answer: Thank you for your suggestion. Firstly, we have added relevant references (for details see line 61). Secondly, according to the suggestions of reviewer, a section is added to the introduction regarding the methods used to prepare bioactive peptides as well as the parameters affecting their bioactivity (for details see line 63-67).
Why did you select these enzymes (Papain, neutrase, flavourzyme and alcalase) for your research? Was there a specific reason?
Answer: Firstly, referring to previous studies, papain, alcalase, flavourzyme and neutrase are frequently used in the production of ACE inhibitory peptides by enzymolysis proteins. different proteases have different restriction sites, peptides of different types, sizes and biological activities were produced. Papain shows wide substrate specificity., the hydrolysis site of alcalase is the C-terminal of hydrophobic amino acid, but the hydrolysis site of flavourzyme and neutrase were not specific. Therefore, in this study, four enzymes were compared to select the best hydrolytic enzyme for the production of ACE inhibitory peptide (line 88-93).
L80-88: Is this method of hydrolysate production the best method? Because we know that the activities of these enzymes are different and we can predict the results. I think that it was better to work based on the degree of hydrolysis (DH) instead of the time of hydrolysis. Also, I think that the DH determination can help you for a better discussion.
Answer: According to the opinion of reviewer, we had supplemented the determination of the hydrolysis degree (for details see line 101-103, 202-206).
L161-164: What were these temperatures to simulate?
The discussion part is generally weak. Only results have been reported. No comparison has been made with the results of other studies at all. Therefore, the discussion part of the manuscript should be improved.
Answer: According to the opinion of reviewer, the discussion part of the manuscript had been improved (for details see line 223-226, 281-283, 288-289, 318-322, 332-336). Moreover, The simulated temperature is 37℃(line 183).
L349: Table caption: different letters in the same row or column?
Answer: Thank you for your suggestion. Different letters in the same line indicate significant differences at p < 0.05 (line 372).
L359-360: in figure 2 caption the F1-F5 should be defined.
Answer: Thank you for your suggestion. F1-F5 has been labeled in Figure 2.

Round 2
Reviewer 2 Report
The comments are well applied and the manuscript can be published.
Author Response
Dear Editors and Reviewers:
Thank you for your letter and for the reviewers' comments concerning our manuscript entitled " Purification and Identification of Novel Angioten-sin-Converting Enzyme Inhibitory Peptide from Enzymatic Hydrolysate of Lepidotrigla microptera" (ID: Foods (ISSN 2304-8158). Those comments are all valuable and very helpful for revising and improving our paper, as well as the important guiding significance to our researches. We have studied comments carefully and have made correction which we hope meet with approval. Revised portion are marked in red in the paper.
If you have any question about this paper, please don't hesitate to let me know.
Best regards.
Sincerely yours,
Hu Xuejia
